# Cellular miR-6741-5p as a Prognostic Biomarker Predicting Length of Hospital Stay among COVID-19 Patients

**DOI:** 10.3390/v14122681

**Published:** 2022-11-30

**Authors:** Shaw M. Akula, John F. Williams, Lok R. Pokhrel, Anais N. Bauer, Smit Rajput, Paul P. Cook

**Affiliations:** 1Department of Microbiology & Immunology, Brody School of Medicine, East Carolina University, Greenville, NC 27834, USA; 2Department of Internal Medicine, Brody School of Medicine, East Carolina University, Greenville, NC 27834, USA; 3Department of Public Health, Brody School of Medicine, East Carolina University, Greenville, NC 27834, USA

**Keywords:** SARS-CoV-2, miRNA, biomarker, COVID-19, LOS

## Abstract

Wide variability exists with host response to SARS-CoV-2 infection among individuals. Circulatory micro RNAs (miRNAs) are being recognized as promising biomarkers for complex traits, including viral pathogenesis. We hypothesized that circulatory miRNAs at 48 h post hospitalization may predict the length of stay (LOS) and prognosis of COVID-19 patients. Plasma miRNA levels were compared between three groups: (i) healthy volunteers (C); (ii) COVID-19 patients treated with remdesivir (an antiviral) plus dexamethasone (a glucocorticoid) (with or without baricitinib, a Janus kinase inhibitor) on the day of hospitalization (I); and COVID-19 patients at 48 h post treatment (T). Results showed that circulatory miR-6741-5p expression levels were significantly different between groups C and I (*p* < 0.0000001); I and T (*p* < 0.0000001); and C and T (*p* = 0.001). Our ANOVA model estimated that all patients with less than 12.42 Log2 CPM had a short LOS, or a good prognosis, whereas all patients with over 12.42 Log2 CPM had a long LOS, or a poor prognosis. In sum, we show that circulatory miR-6741-5p may serve as a prognostic biomarker effectively predicting mortality risk and LOS of hospitalized COVID-19 patients.

## 1. Introduction

Severe acute respiratory syndrome coronavirus 2 (SARS-CoV-2) causes Coronavirus Disease 2019 (COVID-19) [1]. As of 2 September 2022, over 604 million cases of COVID-19 have been documented, with death toll surpassing 6.5 million globally (WHO report). The health and economic toll of COVID-19 will increase over time. Further, the emergence and spread of SARS-CoV-2 variant of concern (VOC) have raised questions about the efficacy of current vaccines and monoclonal antibody therapies [2].

Early manifestations of SARS-CoV-2 infection are protean, but ageusia and anosmia are common [3]. Other common symptoms include fever, cough, dyspnea, myalgias and/or fatigue [4]. In severe cases, there may be acute respiratory distress syndrome (ARDS) or even multi-organ failure [5]. There are increasing reports of some survivors suffering Long COVID [6], a chronic illness with ongoing multidimensional symptomatology and disability that may last for weeks to years after the initial infection [7]. The most common symptom associated with Long COVID is asthenia [8,9]. The other general symptoms that may be associated with Long COVID are fever, chills, anorexia, and general malaise [10]. Long COVID may affect multiple organs in the cardiovascular [11], respiratory [12], nervous [13], digestive [14], urinary [15], and the reproductive systems [16]; and thus may result in a varied tissue-specific clinical manifestations.

MicroRNAs (miRNAs) are small non-coding RNAs (ncRNA), 19–25 nucleotides in length, derived from hairpin shaped precursor molecules encoded by the genomes of animals, plants, and viruses [17]. miRNAs control gene expression and regulate a wide array of biological processes by targeting messenger RNAs (mRNA) and inducing translational repression or accelerated RNA degradation [18]. The miRNA expression profiles are tissue-/cell-specific [19]. In recent years, miRNAs have been studied as likely candidates involved in most biologic processes; they have been implicated in many human diseases, including COVID-19 [20]. The circulatory miRNAs present in blood are enclosed in vesicles (exosomes), which protect them from degradation [21]. The stability of miRNAs in blood, their tissue specificity, and the ease with which they can be quantified make them a viable alternate to serve as a clinical biomarker for a wide range of diseases. We hypothesized that the dynamic changes in the circulatory miRNAs could reflect the severity of COVID-19 and thereby better predict length of hospital stay (LOS) and mortality risk. Our study demonstrates that miR-6741-5p plasma levels could predict LOS of hospital admitted patients by 48 h with high confidence and mortality risk. The study further describes the significance of miR-6741-5p in the biology of COVID-19.

Treating COVID-19 and associated respiratory illness can be a major strain on the healthcare and economic systems. Pandemics may result in a sudden increase in demand for inpatient facilities with a corresponding increase in the number of infected cases [22]. Liquid biomarkers such as miR-6741-5p have the advantage of real-time, non-invasive longitudinal monitoring of a disease condition and heterogeneity over tissue markers [23]. This is the first study that describes a miRNA with potential to serve as a prognostic biomarker for predicting LOS and mortality risk. Predicting factors (such as circulating miR-6741-5p) associated with the LOS can not only aid in triaging patients in a timely manner but also assist in efficiently treating patients and managing hospitals.

## 2. Materials and Methods

### 2.1. Human Participants and Samples

This was a single-center study. The patient plasma samples were obtained from the Department of Pathology, Vidant Medical Center, Greenville, NC. These specimens were from COVID-19 positive patients with moderate-severe disease. The patient details and the demographics are provided in Table 1 and Table 2, respectively. The inclusion criteria for the COVID-19 participants were, (i) age ≥ 18 years; (ii) hospitalized ≤ 48 h; (iii) confirmed SARS-CoV-2 by PCR; and (iv) moderate-severe pneumonia as assessed by radiographic imaging and oxygen requirements. All subjects were requiring supplemental oxygen at the time of the blood sampling. Plasma from healthy volunteers were used as controls. The healthy volunteers were apparently healthy people who did not exhibit symptoms of COVID-19 or any other form of respiratory distress in the past 14 days. None of them were vaccinated for COVID-19 as it was not available at that time. We age-matched the healthy controls by choosing healthy volunteers aged between 30 and 66 years of age. Plasma samples were collected from the hospitalized participants at two different time points: on the day of admission (treated with an antiviral plus 6 mg of dexamethasone administered orally) and 48 h post treatment. Treatment of COVID-19 patients involved an antiviral, remdesivir and dexamethasone (a glucocorticoid). Six of the eighteen subjects received baricitinib, an oral Janus kinase inhibitor with potent anti-inflammatory effects, in addition to dexamethasone. Different antivirals were used, which are listed in Table 1. A one-time blood draw from healthy volunteers provided plasma samples that were used as controls in this study.

### 2.2. Study Design and Approval

COVID-19 positive human plasma samples were obtained from the department of Pathology, East Carolina University. Control plasma samples were obtained from healthy participants with informed written consent before inclusion in the study in accordance with Declaration of Helsinki principles. All protocols were approved by the University and Medical Center IRB (UMCIRB) review board (approved UMCIRB study number is UMCIRB 20-001604).

### 2.3. Whole Transcriptome Assay

We determined the circulating miRNA profile in the human plasma specimens by performing the HTG EdgeSeq miRNA Whole Transcriptome Assay (miRNA WTA) (HTG Molecular Diagnostics, Inc. Tucson, AZ, USA). The HTG EdgeSeq miRNA WTA enables us to measure the expression of 2083 human miRNA transcripts using next-generation sequencing (NGS). The miRNA expression profile was determined in 20 μL of plasma samples as described previously in our earlier studies [24]. Batch effects are technical variations between the measurements due to factors, such as sample or reagent batches. We limited the batch effects using the following checklist while performing NGS: samples were randomized in a balanced manner; samples were always used in duplicates; correlation of all sample pairs was confirmed; normalization was performed to transform the data in such a way as to make the data from different replicates and experimental conditions directly comparable; and a multivariate method of analysis (principal component analysis-PCA) performed to detect any possible batch effects. PCA was performed based on the Scree plot (eigen values vs. components).

### 2.4. Dual-Luciferase Reporter Assay

The 3ʹ-UTR (with wild-type and mutant binding sites for miR-6741-5p) of HNRNPA1L2 (accession number: NM_001011724), and APOBEC3H (accession number: FJ376617), respectively, was PCR amplified and then cloned into XhoI/XbaI site located at 3′UTR of pmirGLO dual-luciferase vector (Promega, Madison, WI, USA). The 3′-UTR of HNRNPA1L2 and APOBEC3H with mutations to binding sites (MUT1 and 2) were generated by site directed mutagenesis as per standard protocols [24]. HEK-293T cells were plated onto 6-well plates. At 24 h post-plating, HEK-293T cells were co-transfected with HNRNPA1L2 (or APOBEC3H) 3′-UTR luciferase reporter plasmid and miR-6741-5p mimic (or control mimic) using FuGene HD (Promega). At 48 h post transfection, the Renilla luciferase activity was measured using the dual luciferase reporter assay system (Promega) as per manufacturer’s recommendations. The oligos tested in this study are provided in Appendix A.

### 2.5. Prognosis Evaluation

We performed One-way analysis of variance (ANOVA) to predict good or poor prognosis based on the log2 CPM values for the COVID-19 patients treated with remdesivir and dexamethasone (with or without baricitinib) on the day of hospital admission based on the circulatory miR-6741-5p expression levels (I group), and statistical comparisons were made between the healthy volunteers (C group) vs. I group, I group vs. T group (COVID-19 patients at 48 h post treatment), and C group vs. T group, using Tukey HSD post-hoc test. Our ANOVA model and parameter estimates for the three groups of patients are presented in Table 3, and Table 4 shows the ANOVA model and parameter estimates for predicting good or poor prognosis in COVID-19 patients treated with remdesivir and dexamethasone (with or without baricitinib) on the day of hospital admission based on the circulatory miR-6741-5p expression levels. Parameters were estimated with robust standard errors using HC3 method. Accordingly, the ANOVA model (Table 4) explaining good or poor prognosis is represented by Equation (1).
Good or poor prognosis = β × Log2 CPM + ε_i_(1)
where β denotes coefficient, and ε_i_ denotes standard error representing any variance unaccounted for by the model. Based on our ANOVA model (Table 4), we then determined the estimated marginal mean for log2 CPM, which may serve as a “clinical threshold” between survival and death among COVID-19 patients that received the treatment on the day of hospital admission. We then compared the model predicted prognosis (survival or death) with the LOS and presented in Table 1.

### 2.6. Statistics

The data satisfied normality (K-S test, *p* > 0.1); they were used untransformed unless stated otherwise. One-way analysis of variance (ANOVA) was performed to determine the statistical significance. ANOVA was performed using IBM SPSS v26 (Cary, NC, USA) to determine significant differences between the treatment and control groups, followed by Tukey HSD post-hoc test for multiple comparisons. The *p*-value was established at the 0.05 level.

## 3. Results

### 3.1. Differential Expression of Circulating miRNAs in COVID-19 Patients Treated with Dexamethasone

The clinical details of all patients are summarized in Table 1. By using the extraction-free HTG EdgeSeq system and bioinformatical analyses, the expression of 2083 circulating miRNAs were analyzed in the plasma derived from COVID-19 patients when compared to apparently healthy participants. The three comparisons were between the following groups: (i) Plasma from COVID-19 patients treated with remdesivir plus dexamethasone (with or without baricitinib) on the day of their admission to hospital (group I) versus plasma from healthy volunteers (group C); (ii) Group I versus plasma from COVID-19 patients obtained 48 h post treatment (group T); and Group C versus group T. Plasma samples for group I was collected before treatment on the day of admission.

Principal component analysis (PCA) shows that the miRNA expression profiles in group I and group C; group I and group T; and group C and group T segregated into distinct clusters along principal component 1 and principal component 2, explaining 77.4% and 10%; 84.6% and 5.6%; and 78.8% and 8.3% of the total variances, respectively (Figure 1A–C). The scree plots are provided in Appendix A which depict principal components and their eigenvalues. To identify miRNAs that were significantly differentially expressed between the two groups, volcano plot filtering was performed (Figure 1D–F). The focus of this study was to analyze the differences in the miRNA profiles within groups I and T. The number of differentially expressed miRNAs between groups I and T are relatively less when compared to groups I and C, and groups T and C (Figure 1D–F). The changes in miRNA profile between groups I and C were recently published [24].

A total of nine miRNA levels were altered in a span of 48 h post dexamethasone treatment (Figure 1E and Figure 2). Eight miRNAs were downregulated (Appendix A) while only one miRNA was upregulated in dexamethasone treated COVID-19 infected and hospitalized patients. The upregulated and dexamethasone treatment-induced miRNA was miR-6741-5p (Figure 2A). We focused the rest of our study on miR-6741-5p primarily owing to its dynamic expression pattern.

### 3.2. miR-6741-5p May Serve as a Predictor of Poor Prognosis

The circulatory miR-6741-5p expression was significantly lower in COVID-19 patients (I) compared to healthy volunteers (C) (Figure 2A). COVID-19 patients treated with remdesivir plus dexamethasone (with or without baricitinib) showed a dramatic increase in miR-6741-5p levels (Figure 2A). The miR-6741-5p expression in plasma obtained from patients treated with remdesivir plus dexamethasone (with baricitinib) at 48 h post treatment was slightly higher than those treated without baricitinib (12.82 vs. 10.3 log2 counts per million (CPM)). Plasma miR-6741-5p expression levels among the three groups of patients are shown in Figure 2. ANOVA results showed that circulatory miR-6741-5p expression levels were significantly different: (i) between the healthy volunteers (C) and COVID-19 patients treated with remdesivir plus dexamethasone (with or without baricitinib) on the day of hospital admission (I) (*p* < 0.0000001); (ii) between I and T (COVID-19 patients at 48 h post treatment) (*p* < 0.0000001); and (iii) between C and T (*p* = 0.001) (Figure 2A). ANOVA model and parameter estimates for the three groups of patients are presented in Table 3.

Surprisingly, it was determined that the antiviral plus dexamethasone treatment (with or without baricitinib)-induced sudden surge in the miR-6741-5p levels was detrimental to the subject (Figure 2B). We could accurately predict the LOS for the hospitalized patients within 48 h post treatment (Table 1). Our ANOVA results showed that there was a significant difference in circulatory miR-6741-5p levels between the patients that had good prognosis or a shorter LOS versus that died, had a poor prognosis, or a longer LOS (*p* = 4.68 × 10^−16^) (Figure 2B).

Our ANOVA model, as presented in Table 4, predicting good or poor prognosis in COVID-19 patients treated with remdesivir and dexamethasone (with or without baricitinib) on the day of hospital admission based on the circulatory miR-6741-5p expression levels are presented below (Equations (1) and (2)). The coefficient of determination (R squared) of the model was 0.997 (Table 4), suggestive of good model fit with sufficient predictive power. Parameters were estimated with robust standard errors using HC3 method. Our model estimated marginal mean of 12.42 for log2 CPM may serve as a “clinical threshold” between survival and death.
Good prognosis = 11.168 Log2 CPM + 0.170(2)
Poor prognosis = 13.675 Log2 CPM + 0.130(3)

### 3.3. miR-6741-5p Targets APOBEC3H

One miRNA may regulate multiple genes, while one gene may also be targeted by many miRNAs [25]. Accordingly, miR-6741-5p can possibly target several genes (Appendix A). Using bioinformatics tools (miRDB and TargetScan) [26,27], we proposed to test the top two promising targets to miR-6741-5p. The selection criteria for the two candidates were based on the target score [26] as well as the biological relevance of the target. The plausible binding site of miR-6741-5p is located in the 3′-UTR of HNRNPA1L2 and APOBEC3H mRNA (Appendix A). To confirm the ability of miR-6741-5p to specifically bind and interact with HNRNPA1L2 and APOBEC3H 3′-UTR regions, we used the conventional luciferase reporter assay. Transfection of HEK293T cells with miR-6741-5p mimic significantly lowered the luciferase activity of wild-type (WT) APOBEC3H 3′-UTR vector plasmid and not the vector encoding the WT-HNRNPA1L2 3′-UTR (Figure 3). In the conditions tested, miR-6741-5p mimic did not lower luciferase activity of the vector encoding the WT-HNRNPA1L2 3′-UTR and this could be due to one or more of the following reasons: (i) predictions based on genome-wide computation models may not necessarily result in functional target suppression [28,29]; (ii) limitations of the luciferase-based target validation [30]; and (iii) to the cell-type tested in this assay. Transfection of cells with miR-NC could specifically inhibit luciferase activity of wild-type (WT) APOBEC3H 3′-UTR vector plasmid (Figure 3). The specificity the miR-6741-5p interactions with APOBEC3H 3′-UTR region was confirmed by using MUT1 and MUT2 variants (Appendix A): transfection of HEK293T cells with miR-6741-5p mimic significantly lowered the luciferase activity of MUT1 APOBEC3H 3′-UTR vector plasmid and not the MUT1 APOBEC3H 3′-UTR vector plasmid (Figure 3). Taken together, the results suggest a key role for target site located in 3’-UTR region of APOBEC3H (position 24–31; Appendix A) with which miR-6741-5p interacts.

## 4. Discussion

We hypothesized circulatory miRNAs to provide predictive significance for the prognosis of COVID-19 patients. There are over 5000 publications that describe circulating miRNAs as biomarkers and 977 publications that describe circulating miRNAs as prognostic biomarkers. Interestingly, there are about 26 publications that describe the role of circulating miRNAs in COVID-19 and only one manuscript describes the cardiovascular signatures of severe COVID-19 that could predict mortality [31]. The study determined a combination of circulatory miRNAs, inflammatory and endothelial cell biomarkers to serve as predictors of mortality in COVID-19 patients with severe illness.

The goal of this study was to determine the minor differences observed in the circulatory miRNA profiles of moderate to severe COVID-19 patients on the day of hospitalization (I) when they received remdesivir plus dexamethasone (with or without baricitinib) treatment and at 48 h post treatment (T). The difference in changes to miRNA profile within 48 h post treatment was assumed to be the net effect of antiviral plus dexamethasone with or without baricitinib treatment. PCA is used to identify strong patterns in large, complex data sets [32]. It is considered as the initial step in an exploratory analysis on expression data such as miRNA profiles [33]. This is widely used effectively to reduce a large set of variables into smaller, easier-to-analyze sets while retaining meaningful information [34]. We found PCA of plasma miRNA profiles could distinguish COVID-19 patients (group I) from healthy volunteers (group C). PCA analysis showed distinct clusters for group C and I (Figure 1A,C); whereas COVID-19 patients (group I) clustered relatively closer to COVID-19 patients treated for 48 h (group T) (Figure 1B). Such a cluster pattern between groups I and T is partly because of the shorter time interval post treatment and the inherent variations in the host responses to the treatment. It is evident that the discriminating miRNAs, such as miR-6741-5p, that result from such an analysis may prove to be robust biomarkers.

Only nine circulatory miRNAs were differentially expressed between groups I and T compared to groups I and C, and groups T and C (Figure 1D–F). Of these, only miR-6741-5p was upregulated in COVID-19 patients that were treated (T) (Figure 2A). We were interested in the expression of miR-6741-5p for two reasons: (i) the significant difference in the expression levels of miR-6741-5p between the three groups of participants (Figure 2A); and (ii) the putative targets of miR-6741-5p and their possible relevance to COVID-19 biology (Figure 3).

There was a unique pattern of expression in COVID-19 patients who received treatment. SARS-CoV-2 infection as such lowers the expression of miR-6741-5p (Figure 2A). Upon treatment, in a subset of COVID-19 patients we observed a sudden spike in the miR-6741-5p expression (Figure 2B) and that was a sign of poor prognosis, or longer LOS (Table 1). The COVID-19 patients who survived post treatment generally had only a slight or no change in the expression of miR-6741-5p (Figure 2B). An estimated marginal mean of 12.42 for Log2 CPM (shown as dotted line in Figure 2B) may serve as a “clinical threshold” that may allow clinicians to determine who may survive (good prognosis) or die (poor prognosis) among COVID-19 patients. More specifically, all patients with below 12.42 Log2 CPM survived, had a short LOS, or a good prognosis (*n* = 7); whereas all patients with over 12.42 Log2 CPM died, had a long LOS, or a poor prognosis (*n* = 11). The LOS for all patients are shown in Table 1. Inflammatory biomarkers such as C-reactive protein (CRP), D-dimer, and ferritin are often elevated in COVID-19 patients [35]. We did not observe a significant correlation between the levels of inflammatory biomarkers during early times of treatment and the treatment outcome as suggested in earlier studies [24,36]. However, medical interventions such as mechanical ventilation and oxygenation seem to have a correlation with the prognosis of the disease. All the individuals who were on ventilation and/or received oxygen for an extended period of time had poor prognosis or died (Appendix A). The miR-6741-5p expression in plasma obtained from patients treated with remdesivir plus dexamethasone (with baricitinib) at 48 h post treatment was slightly higher than those treated without baricitinib (12.82 vs. 10.3 log2 CPM). This can be explained by the fact that there are still concerns that JAK inhibitors such as baricitinib may increase the incidence of COVID-19-associated pulmonary aspergillosis (CAPA) when used with corticosteroids [37].

In our recently concluded study [24], we did not mention miR-6741-5p as the discussion was only on the select miRNAs that were altered ≥10 folds due to SARS-CoV-2 infection. The effects of miR-6741-5p on COVID-19 biology could be multiple based on the potential target genes (Appendix A). The overall pleiotropic effects of a sudden increase in miR-6741-5p expression in COVID-19 patients seem to lower adaptive immunity, pro-inflammatory response, healing process, cell proliferation and survivability while increasing viral replication; all of which may result in a poor prognosis (Appendix A) [38,39,40,41,42,43,44,45,46]. It is important to note that the miR-6741-5p expression levels in those with poor prognosis are at comparable levels to what is observed in healthy individuals (Appendix A). This is a scientific paradox. Herein, we will attempt to solve the paradox using our working hypothetical model as follows: the biological effect of a miRNA is going to be different in a healthy individual compared to an individual who is responding to a viral infection; the biotic milieu is different (Figure 4). In a healthy naïve individual, an innate immune response is only triggered following an exposure to a pathogen and so there is not going to be an adaptive immune response [47]. The inflammatory responses and cell proliferation in a healthy individual are at adequate levels. When SARS-CoV-2 infects a person with a good prognosis, there is a decrease in the miR-6741-5p levels that promote an innate immune response followed by a robust specific adaptive immune response to fight the viral pathogen. There is an increase in pro-inflammatory response to combat virus-induced pathology and inflammation, followed by healing process (cell proliferation and survivability). In a subset of few patients where there is a sudden surge in the miR-6741-5p levels (comparable to healthy individuals), the net effect is different as they are affected with an ensuing virus-induced pathology. Such a sudden surge in miR-6741-5p leads to a decrease in adaptive viral specific immunity, pro-inflammatory response, antiviral response, cell proliferation and survivability; and thus a poor prognosis. In this study, we were also able to establish miR-6741-5p to physically bind, interact, and regulate expression of APOBEC3H gene (Figure 3). Human APOBEC3H has two major functions: DNA cytosine deaminase [48] and DNA editing enzymatic activity [49]. The gene ontology (GO) annotations for APOBEC3H are provided in Appendix A. The DNA editing enzyme function of APOBEC3H plays a crucial role in innate and adaptive immune responses to viruses like herpesviruses, retroviruses, and potentially RNA viruses such as SARS-CoV-2 [39,42,50]. The APOBEC3H is regulated by interferons (another key mediator of innate immunity) [51] and thus may not have a direct bearing in healthy individuals when compared to infected patients. An integrated network and dynamical reasoning assembler (INDRA)-interactive pathway map (INDRA-IPM) [52] representing the ability APOBEC3H to negatively regulate viral genome is presented in Appendix A. In a recently concluded study, it was determined that the levels of APOBEC3H were higher in older individuals [53]. Controneo et al. [53] concluded the necessity to understand if APOBEC3H isoforms affect prognosis in older COVID-19 patients.

## 5. Limitations

Besides the sample size, there are other limitations of this study. First, the samples were from patients hospitalized with COVID-19 in the summer of 2021. At that time, the Delta variant was the predominant strain of SARS-CoV-2 circulating in the US and in our hospital. Whether the same miRNAs are expressed with Omicron and other variants remain to be determined. We plan to investigate this thesis using plasma samples from patients hospitalized with COVID-19 this year when Omicron has been responsible for >95% of hospitalizations. Also, we do not know what effects, if any, other antivirals such as molnupiravir or nirmatrelvir/ritonavir might have on the miRNA profiles. Finally, all patients in this study presented at various times in the illness. As such, we do not have a clear picture of when the drop in the level of miR-6741-5p relative to infection occurs. Despite these limitations, we are confident that this may serve as a steppingstone to understand the role of miR-6741-5p-regulated APOBEC3H in COVID-19 biology.

## 6. Conclusions

Based on the above available data, we propose a model (Figure 4) to explain the role of miR-6741-5p in COVID-19 biology post initiation of treatment. Each individual responds in a unique way to treatment. In healthy control participants (C), a basal level of miR-6741-5p is observed. SARS-CoV-2 infection results in a sharp decline in miR-6741-5p expression. A sharp decline in miR-6741-5p results in an upregulation of APOBEC3H-induced antiviral response. Such a virus-induced 10-fold APOBEC3H upregulation has been recorded in earlier studies [54]. Treatment of these COVID-19 patients results in two scenarios: (i) a sharp spike of miR-6741-5p; or (ii) a modest to no change in miR-6741-5p expression. A sharp spike in the miR-6741-5p expression results in a sudden depletion of APOBEC3H-induced antiviral activity that may support an increase in SARS-CoV-2-induced pathogenesis resulting in poor prognosis and vice versa. In sum, we show that circulatory miR-6741-5p may serve as a prognostic biomarker effectively predicting mortality risk and LOS of hospitalized COVID-19 patients.

Future studies are aimed at testing (i) our statistical model in a large sample population; and (ii) if miR-6741-5p expression can determine the prognosis at the time of admission. The APOBEC3H antiviral activity depends on recognition of RNA or DNA strand [55]. Accordingly, the direct target of APOBEC3H in the SARS-CoV-2 genome is underway.

## Figures and Tables

**Figure 1 viruses-14-02681-f001:**
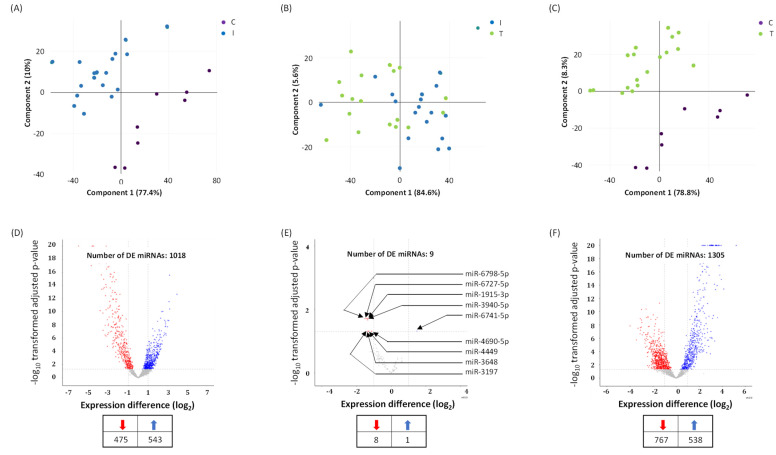
Differentially expressed (DE) miRNAs in COVID-19 patients with moderate-severe disease (I) compared to treated (T) and healthy participants (C). (**A**–**C**) Principal component analysis (PCA) plot showing transcriptome differences between miRNA expression in I, T, and C group of COVID-19 patients. Each dot represents one of the total number of samples (12 samples from I and T group of COVID-19 patients and 8 samples from healthy participants) used in the sequencing-based screening. (**D**,**E**) Volcano plot for differentially expressed miRNAs, showing distribution of significance [−log10(*p*-value)] vs. fold change [log2(fold change)] for all genes. The red and blue dots represent downregulated and upregulated miRNA expression in I vs. C groups (**D**); I vs. T groups (**E**); and T vs. C groups (**F**), respectively. C denotes healthy volunteers, I denotes COVID-19 patients treated with remdesivir (an antiviral) plus dexamethasone (with or without baricitinib, a Janus kinase inhibitor) on the day of hospitalization, and T denotes COVID-19 patients at 48 h post treatment. The number of DE miRNAs is listed in the table provided under panels D, E, and F. Arrows denote specific miRNAs.

**Figure 2 viruses-14-02681-f002:**
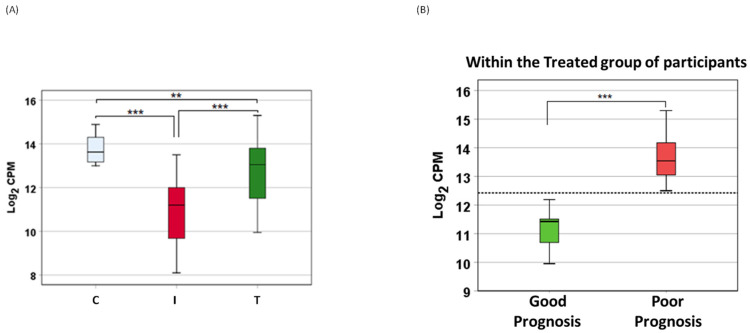
**Differential expression of miR-6741-5p.** (**A**) Box plot showing variation in miR-6741-5p expression (log2 CPM) among different patient groups: healthy volunteers (C), patients with COVID-19 (I), and patients with COVID-19 that received antiviral plus dexamethasone treatment (T). ‘**’ denotes significant difference between the means at *p* = 0.001; and ‘***’ denotes significant difference between the means at *p* < 0.0000001. Statistical significance was established at 0.05 level using ANOVA followed by Dunnett T3 post-hoc test. (**B**) Box plot showing variation in miR-6741-5p expression (log2 CPM) among the patients with COVID-19 that received treatment and either survived or died. ‘***’ denotes significant difference between the means at *p* = 4.68 × 10^−16^ using ANOVA. The horizontal dotted line denotes estimated marginal mean of 12.42, which may serve as a “clinical threshold” between survival and death.

**Figure 3 viruses-14-02681-f003:**
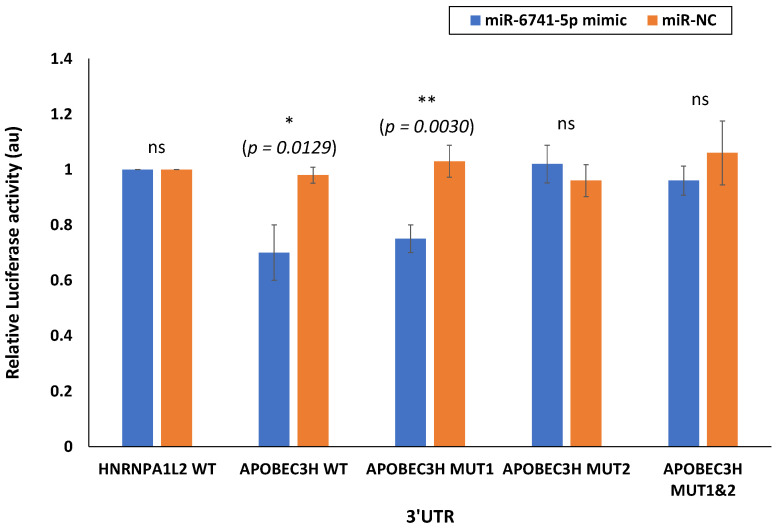
miR-6741-5p specifically binds and interact with APOBEC3H. Luciferase activity in HEK293T cells transfected with pmirGLO dual-luciferase vector with 3′UTR of HNRNPA1L2 (HNRNPA1L2 3′UTR) and APOBEC3H (APOBEC3H 3′UTR). HEK293T cells were transfected with HNRNPA1L2 3′UTR or APOBEC3H 3′UTR, co-transfected with HNRNPA1L2 3′UTR or APOBEC3H 3′UTR and miR-6741-5p mimic, co-transfected with HNRNPA1L2 3′UTR or APOBEC3H 3′UTR and control mimic (miR-NC). We also tested APOBEC3H MUT1 and MUT2 3′UTR in this study. Luciferase activity was monitored at 48 h post-transfection and was normalized against the effect of the miRNAs on the Firefly luciferase reporter alone. The relative luciferase activity for cells transfected with HNRNPA1L2 3′UTR or APOBEC3H 3′UTR is considered as 1 au. The *x*-axis denotes the UTRs transfected, and *y*-axis indicates the relative luciferase activity. Bars represent average ± s.d. of three individual experiments. One-way ANOVA was performed compare between multiple groups. *p* value of 0.05 or less was considered statistically significant. * denotes *p* = 0.0129; ** denotes *p* = 0.0030; ‘ns’ denotes not significant at *p* = 0.05.

**Figure 4 viruses-14-02681-f004:**
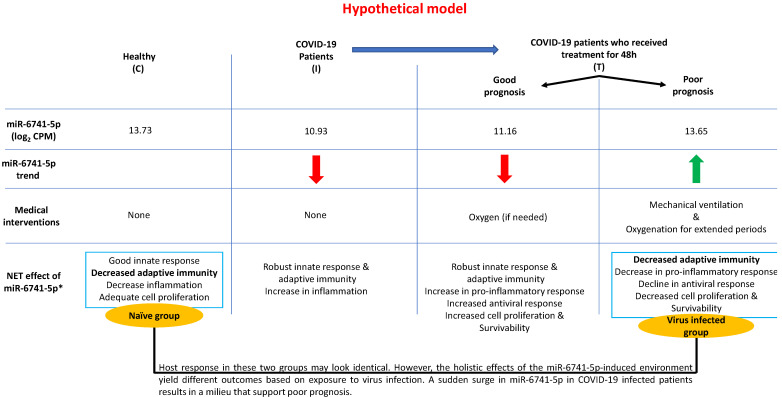
A schematic model describing the working hypothesis. * refer to Appendix A.

**Table 1 viruses-14-02681-t001:** Our predictions based on expression values for miR-6741-5p.

				Log_2_ CPM		
Patient ID	Age	Treatment Received	Co-Morbidities	Day 0	Day 02	Prediction-Based on CPM Values	LOS
A	75	Remdesivir + dexamethasone	HTN, paraplegia	9.5	11.5	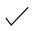	Discharged on day 4
B	73	Remdesivir + dexamethasone	HTN, DM, obesity	10.4	11.5	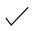	Discharged on day 4
C	68	Remdesivir + dexamethasone	HTN, asthma, DM, obesity	8.9	10.7	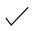	Discharged on day 5
D	77	Remdesivir + dexamethasone	DM	11.4	13.2	X	Dead on day 16
E	49	Remdesivir + dexamethasone	Renal transplant, HTN, DM	11.0	13.4	X	Discharged alive; died 3 days later
F	71	Remdesivir + dexamethasone	HTN, DM, prostate cancer	12.1	13.8	X	Discharged on day 23
G	56	Remdesivir + dexamethasone	ESRD, HTN, DM	8.2	9.9	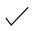	Discharged on day 12
H	73	Remdesivir + dexamethasone	HTN, COPD	11.2	13.4	X	Dead on day 9
I	45	Remdesivir + dexamethasone	DM, obesity	8.5	10.2	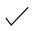	Discharged on day 5
J	72	Remdesivir + dexamethasone	HTN, COPD, paraplegia	9.7	12.2	X	Dead on day 15
K	77	Remdesivir + dexamethasone	HTN, CHF, cirrhosis	8.6	12.6	X	Dead on day 8
L	67	Remdesivir + dexamethasone	HTN, DM, obesity	11.2	12.9	X	Dead on day 26
M	66	Dexamethasone + baricitinib	HTN	11.0	11.3	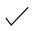	Discharged on day 6
N	66	Dexamethasone + baricitinib	DM, obesity, sickle cell trait	13.2	13.8	X	Discharged on day 57
O	68	Dexamethasone + baricitinib	HTN, DM, obesity	13.0	15.3	X	Died on day 16
P	75	Dexamethasone + baricitinib	No comorbid conditions	12.0	12.8	X	Discharged on day 15
Q	57	Dexamethasone + baricitinib	HTN, DM, obesity	13.1	14.8	X	Discharged on day 21
R	64	Dexamethasone + baricitinib	CHF, CAD	9.0	10.1	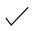	Discharged on day 5

LEGEND: √ Prognosis is good; X Prognosis is bad; CPM denotes count per million.

**Table 2 viruses-14-02681-t002:** Sample information.

Parameters	COVID-19 Patients (*n* = 12)	Healthy Volunteers (*n* = 8)
Age, y, mean (SD)	66.6 (18)	46 (7.3)
Sex, *n* (%)		
Male	8 (44.4%)	3 (37.5%)
Female	10 (55.5%)	5 (62.5%)
Sample collected	Blood (plasma)	Blood (plasma)

**Table 3 viruses-14-02681-t003:** Results of Univariate Analysis of Variance (ANOVA) among the three groups of patients and parameter estimates using robust standard errors (HC3 method). The coefficient of determination (R squared) of the model was 0.988, suggestive of good model fit with sufficient predictive power.

**Tests of Between-Subjects Effects**						
**Dependent Variable: Log_2_ CPM**						
**Source**	**Type III Sum of Squares**	**df**	**Mean Square**	**F**	**Sig.**	
**Model**	19,528.148 ^a^	3	6509.383	3593.522	7.96 × 10^−122^	
**Group**	19,528.148	3	6509.383	3593.522	7.96 × 10^−122^	
**Error**	228.239	126	1.811			
**Total**	19,756.387	129				
**Parameter**	**B**	**Robust Std. Error**	**t**	**Sig.**	**95% Confidence Interval**	
					**Lower Bound**	**Upper Bound**
**Control (C)**	13.733	0.129	106.593	2.40 × 10^−125^	13.478	13.988
**Infected (I)**	10.936	0.214	51.009	5.33 × 10^−86^	10.512	11.361
**Infected + Treated (T)**	12.839	0.193	66.556	4.91 × 10^−100^	12.457	13.221

^a^ R Squared = 0.988 (Adjusted R Squared = 0.988); Computed using alpha = 0.05.

**Table 4 viruses-14-02681-t004:** Results of Univariate Analysis of Variance (ANOVA) among patients infected with SARS-CoV-2 that received dexamethasone treatment for 48 h (with good and poor prognosis). The coefficient of determination (R squared) of the model was 0.997, suggestive of good model fit with sufficient predictive power.

**Source**	**Type III Sum of Squares**	**df**	**Mean Square**	**F**	**Sig.**	
**Model**	8977.149	2	4488.575	8020.432	1.75 × 10^−65^	
**Good prognosis**	8977.149	2	4488.575	8020.432	1.75 × 10^−65^	
**Error**	29.101	52	0.560			
**Total**	9006.251	54				
**Parameter**	**B**	**Robust Std. Error**	**t**	**Sig.**	**95% Confidence Interval**	
					**Lower Bound**	**Upper Bound**
**Poor prognosis**	13.675	0.130	104.989	3.20	13.414	13.936
**Good prognosis**	11.168	0.170	65.825	9.21 × 10^−52^	10.827	11.508

R Squared = 0.997 (Adjusted R Squared = 0.997); Computed using alpha = 0.05.

## Data Availability

Data are contained within the article or Appendix A.

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
