# Peer review of "Cellular miR-6741-5p as a Prognostic Biomarker Predicting Length of Hospital Stay among COVID-19 Patients"

_viruses, 2022, doi:10.3390/v14122681_

Round 1
Reviewer 1 Report
The model presented in the paper is interesting but unfortunately, in my opinion, the sample size is not significant to draw significant clinical conclusions - and - moreover the authors did not considered confounding factors, such as co-morbidities etc.
The model should be also validated in an independent dataset where the discovery of diff expressed Myrna has not been done.
minor.
Fig1(E) caption: I guess should be I vs T .
Reviewer 2 Report
As the title suggests, the results clearly suggest that miR-6741-5p is an useful biomarker. There was one formatting error on page 2, line 66. Other than that, it is well organized.
Reviewer 3 Report
Akula et al. obtained patient plasma miRNA samples from COVID-19 positive patients with moderate-severe disease. The miRNA levels were compared between three groups: (i) healthy volunteers; (ii) COVID-19 patients treated with remdesivir (an antiviral) plus dexamethasone (a glucocorticoid) (with or without baricitinib, a Janus kinase inhibitor) on the day of hospitalization; (iii); and T (COVID-19 patients at 48h post treatment). The authors showed that circulatory miR-6741-5p expression levels were significantly different between groups C and I (p<0.0000001); I and T (p<0.0000001); and C and T (p=0.001). Patients with less than 12.42 Log2 CPM had a short LOS, or a good prognosis, whereas all patients with over 12.42 Log2 CPM had a long LOS, or a poor prognosis. Circulatory miR-6741-5p may serve as a prognostic biomarker effectively predicting mortality risk and LOS of hospitalized COVID-19 patients.
The following suggestion for improving and strengthening the manuscript before publication
· I encourage the authors to draw a scree plot in Figure 1 (A-C) along with PCA plots as it would provide a clearer picture of principal components and their variation.
· The number of up and downregulated miRNAs is not clearly mentioned in Figure 1 (D-F). Please mention in the figure how many miRNAs were down and upregulated. Also, what does red and blue dots signify? Please mention
· What parameters were used for screening mRNAs interacting with miR-6741-5p from miRDB and TargetScan? Please mention.
· The authors inferred good or bad prognosis/LOS based on log CPM levels. Please mention how?
· It would be nice if the authors could perform pathway and functional enrichment analysis of final miRNA and mRNA to infer their role in pathways/GO terms. This would even strengthen their findings.
· What does ns stand for in Figure 3?
· The significance of the study needs to be discussed briefly in the Introduction Section.
· The methods section does not mention any hint of prognosis evaluation, but results clarify this in detail. The methods section needs to be properly framed as per the results as it is not well framed.
· Quality of Figures 1 and 2 needs to be improved as legends are not clearly visible.
· Student t-test or One-way analysis of variance (ANOVA) was performed to determine the statistical significance. Please explain what this sentence means in the methods section. Which test did the authors employ?
· How were the batch effects corrected by the authors while collecting miRNA transcripts using next-generation se-132 sequencing (NGS)? Please explain.
Round 2
Reviewer 1 Report
from the comments even I have concerns related to the significance of the study. But as Im not expert in such kind of study I suppose that even limited results may be of interest for the community.
Reviewer 3 Report
Manuscript improved, I am recommending for publication.